# Tracking SARS-CoV-2 variants through pandemic waves using RT-PCR testing in low-resource settings

Asghar Nasir[1], Uzma Bashir Aamir[2], Akbar Kanji[1], Ali Raza Bukhari[1], Zeeshan Ansar[1], Najia Karim Ghanchi[1], Kiran Iqbal Masood[1], Azra Samreen[1], Nazneen Islam[1], Samina Ghani[1], M. Asif Syed[3], Mansoor Wassan[3], Syed Faisal Mahmood[4], Zahra Hasan[1]*

1 Department of Pathology and Laboratory Medicine, Aga Khan University, Karachi, Pakistan, 2 World Health Organization, Islamabad, Pakistan, 3 Department of Health, Government of Sindh, Karachi, Pakistan, 4 Department of Medicine, The Aga Khan University, Karachi, Pakistan

* zahra.hasan@aku.edu

**Data Availability Statement:** All data can be found in the manuscript and supporting information files.

**Funding:** This work was support by the Aga Khan University, Pakistan (University Research Council

## Abstract

COVID-19 resulted in extensive morbidity and mortality worldwide. SARS-CoV-2 evolved rapidly, with increasing transmission due to Variants of Concern (VOC). Identifying VOC became important but genome submissions from low-middle income countries (LMIC) remained low leading to gaps in genomic epidemiology. We demonstrate the use of a specific mutation RT-PCR based approach to identify VOC in SARS-CoV-2 positive samples through the pandemic in Pakistan. We selected 2150 SARS-CoV-2 PCR positive respiratory specimens tested between April 2021 and February 2022, at the Aga Khan University Hospital Clinical Laboratories, Karachi, Pakistan. Commercially available RT-PCR assays were used as required for mutations in Spike protein (N501Y, A570D, E484K, K417N, L452R, P681R and deletion69_70) to identify Alpha, Beta, Gamma, Delta, and Omicron variants respectively. Three pandemic waves associated with Alpha, Delta and Omicron occurred during the study period. Of the samples screened, VOC were identified in 81.7% of cases comprising mainly; Delta (37.2%), Alpha (29.8%) and Omicron (17.1%) variants. During 2021, Alpha variants were predominant in April and May; Beta and Gamma variants emerged in May and peaked in June; the Delta variant peaked in July and remained predominant until November. Omicron (BA.1) emerged in December 2021 and remained predominant until February 2022. The CT values of Alpha, Beta, Gamma and Delta were all significantly higher than that of Omicron variants (p<0.0001). We observed VOC through the pandemic waves using spike mutation specific RT-PCR assays. We show the spike mutation specific RT-PCR assay is a rapid, low-cost and adaptable for the identification of VOC as an adjunct approach to NGS to effectively inform the public health response. Further, by associating the VOC with CT values of its diagnostic PCR we gain information regarding the viral load of samples and therefore the level of transmission and disease severity in the population.

Grant awarded to ZH), the Higher Education Commission, Pakistan (Rapid Research Grant number 236 awarded to ZH) and the World Health Organization, Pakistan (grant awarded to ZH). The funders had no role in study design, data collection and analysis, decision to publish, or preparation of the manuscript.

**Competing interests:** The authors have declared that no competing interests exist.

## Introduction

SARS-CoV-2 is a coronavirus that originated in Wuhan, China in December 2019 and spread globally causing the disease associated with SARS-CoV-2 (COVID-19). According to John Hopkins Coronavirus Resource Center (https://github.com/CSSEGISandData/COVID-19), as of 10 March 2023, SARS-CoV-2 related infections have surpassed over 765 million cases, with 6.71 million deaths globally. Genomic epidemiology of SARS-CoV-2 has been an essential component of monitoring and management of COVID-19 at the public health level. As SARS-CoV-2 strains evolved, it became evident that the virus rapidly acquired mutations particularly in the spike glycoprotein on the virus coat, and that this impacted entry into host cells [1]. Further, as the pandemic surged across the globe, genomic sequencing revealed differences between SARS-CoV-2 variants subsequently identified by specific spike protein mutations, which increased transmissibility and infectivity of strains [2]. It became imperative to track and trace SARS-CoV-2 variants across the global to understand the transmission dynamics of COVID-19.

Centers for Disease Control and Prevention (https://www.cdc.gov/coronavirus/2019-ncov/variants/variant-info) identified variants of concern (VOC) to be characterized by unique spike protein signature mutations, the first four of which were Alpha (B.1.1.7), Beta (B.1.351), Gamma (P.1) and Delta (B.1.617.1/2) lineages. VOC were more transmissible than wild-type SARS-CoV-2 strains and lead to new pandemic surges related to factors including, reduced neutralization by antibodies from previous infection or vaccination [3, 4]. From mid-2021 to November 2021, the Delta variant (originating in India) emerged as the globally dominant variant. In November 2021, a new variant emerged in Botswana, Africa and the World Health Organization (WHO) it as Omicron B.1.1.529 [5]. The Omicron variant subsequently surpassed the Delta variant as a globally dominant variant, as it was highly successful at evading host immune and antibody responses.

To date (1 April 2023), Pakistan with a population of 220 million, has reported an estimated 1.57 million COVID-19 cases from 30.6 million SARS-CoV-2 PCR tests (https://ourworldindata.org/coronavirus-testing). Five successive COVID-19 waves were documented between 2020 and 2022 occurring between; March and July 2020, October 2020 and January 2021, April and May 2021, July and September 2021 and, between December 2021 and February 2022 [1, 2]. Sindh province accounted for 37% of COVID-19 cases in Pakistan (https://covid.gov.pk/stats/pakistan). 81% of COVID-19 cases across Sindh were from Karachi, with a population of 20 million (https://www.sindhhealth.gov.pk/daily_situation_report). There was a need to have an efficient method to identify and trace VOC in the region.

Whole genome sequencing (WGS) is the gold standard for surveillance in an epidemic as it can fully identify known and unknown variant lineages. Sequencing of SARS-CoV-2 strains had identified the introduction of VOC in Pakistan [6–8]. However, next-generation sequencing (NGS) technologies are costly, require complex technical instrumentation and expertise. Therefore, it is not readily available in resource-limited countries like Pakistan where in particular there was a shortage of reagents through the pandemic period in Pakistan. Alternately, RT-PCR-based detection of lineage-specific target mutations can be used to identify known VOC.

WHO, Pakistan supported the implementation of genomic surveillance for SARS-CoV-2 at major diagnostic laboratories including Aga Khan University Hospital (AKUH) as part of a national consortium coordinated by the National Reference Public Health Laboratory at the National Institute of Health, Islamabad. AKUH has worked with the Department of Health, Government of Sindh, through the COVID-19 pandemic to provide rapid, clinical diagnostic testing for suspected cases. Due to the financial and technical constraints we could not

sufficiently scale up WGS as a primary tool to investigate SARS-CoV-2 variants across the pandemic waves. We describe here a strategy which employed a series of commercially available mutation-specific RT-PCR assays to identify VOC through the pandemic. We report the emergence and spread of the Alpha, Beta, Gamma, Delta, and Omicron variants in Karachi.

## Materials and methods

This study received approval from the Ethical Review Committee, The Aga Khan University (AKU).

### SARS-CoV-2 PCR testing

AKU Hospital (AKUH) Clinical Laboratories are accredited by the College of American Pathologists, USA. Routine testing for SARS-CoV-2 by PCR testing is performed on the cobas SARS-CoV-2 assay (Roche diagnostics).

### Selection of study specimens

Between April 2021 and February 2022, a retrospective review of SARS-CoV-2 specimens was conducted to identify the first 10 specimens reported positive on each previous day. Inclusion criteria for samples was a CT (crossing threshold) value $\leq 35$ as measured by the cobas SARS-CoV-2 diagnostic assay. Exclusion criteria, specimens that had a CT value $> 35$, or were duplicate specimens for the same individual.

### PCR assays to identify VOC

RNA was extracted from each SARS-CoV-2 positive nasal swab specimen using the QIAampRNA MiniKit, Qiagen, according to the manufacturer's instructions. RNA was stored at -80˚C until tested further.

S1 Table shows the flow in which the different RT-PCR assays were used. Details of probes/ primers sequences and target gene mutations for each assay are listed in S2 Table. All assays were run on the Quant5 real-time PCR machine, Applied Biosystems, USA.

NovaType SARS-CoV-2, Gold Standard Diagnostics, Eurofins Technologies, identifies N501Y and A570D mutations. It was used to distinguish between Alpha and Beta variants. NovaType II SARS-CoV-2, Gold Standard Diagnostics, Eurofins Technologies identified N501Y, E484K and K417N mutations and was used to distinguish between Alpha, Beta and Gamma variants. Phoenix SARS-CoV-2 Mutant Screen [L452R], Promocure Biotech GmbH, identifies the L452R mutation and was used to identify Delta variants. NovaType Select P681R SARS-CoV-2, Gold Standard Diagnostics, Eurofins Technologies, identifies the P681R mutation and was used to identify Delta variants. GSD NovaType III SARS-CoV-2, Gold Standard Diagnostics, Eurofins Technologies, identifies E484Km L452R and E484Q. It was used to identify Beta/Gamma, Delta or Kappa variants. TaqPath™ COVID-19 CE-IVD RT-PCR, Applied Bio systems identifies the deletion 69–70 in spike protein as S-gene target failure (SGTF). SGTF occured in both Alpha and Omicron (BA.1) variants.

Due to the limited availability of reagents, we were unable to conduct comprehensive lower limit of detection testing for each assay separately. However, we did test them across a range of CT values. We found all six RT-PCR assays the assays to be consistent for samples with CT values up to 35. We focused only on samples of a medium to high viral load for this study, selecting samples with a CT $\leq 35$. The workflow for how each kit was utilized to screen and differentiate the VOC is presented in S1 Fig.

## Results

### SARS-CoV-2 diagnostic testing

We investigated on samples received for SARS-CoV-2 RT-PCR testing at AKUH Clinical Laboratories in Karachi, Sindh. The AKUH received between 17,000 and 32,000 specimens per month for SARS-CoV-2 PCR testing between April 2021 and February 2022. In total, these were 238,324 specimens of which 52,905 (22%) were positive for SARS-CoV-2. Results of specimens tested reported daily to the public health authorities.

Three COVID-19 waves were observed over the study period as shown in Fig 1. Pandemic surges were between April and May 2021, July and September 2021, and December 2021 until February 2022. We investigated the VOC associated with these pandemic waves. Next-generation sequencing supplies were severely restricted due to the increased demands faced by companies, and difficulties in production lines due to lockdowns associated with the pandemic. Therefore, we used a specific mutation RT-PCR based approach to identify VOCs in samples selected to represent the study period.

### Screening for predominant VOCs

The Alpha variant first identified in Pakistan in December 2021 [8], was present at the start of this study in April 2021. By then, RT-PCR kits had been developed for rapid and cost-effective identification of VOCs. However, these remained in limited supply, resulting in delays and modification of testing protocols. The new assay development was in conjunction with the emergence of SARS-CoV-2 VOC. Therefore, we describe here VOC testing based on the RT-PCR assays sequentially available to us between 2021 and 2022, to identify Alpha, then Beta, Gamma, Delta and Omicron variants. The sequence in which the kits were employed is described in S1 Fig, with usage based on their availability through the study.

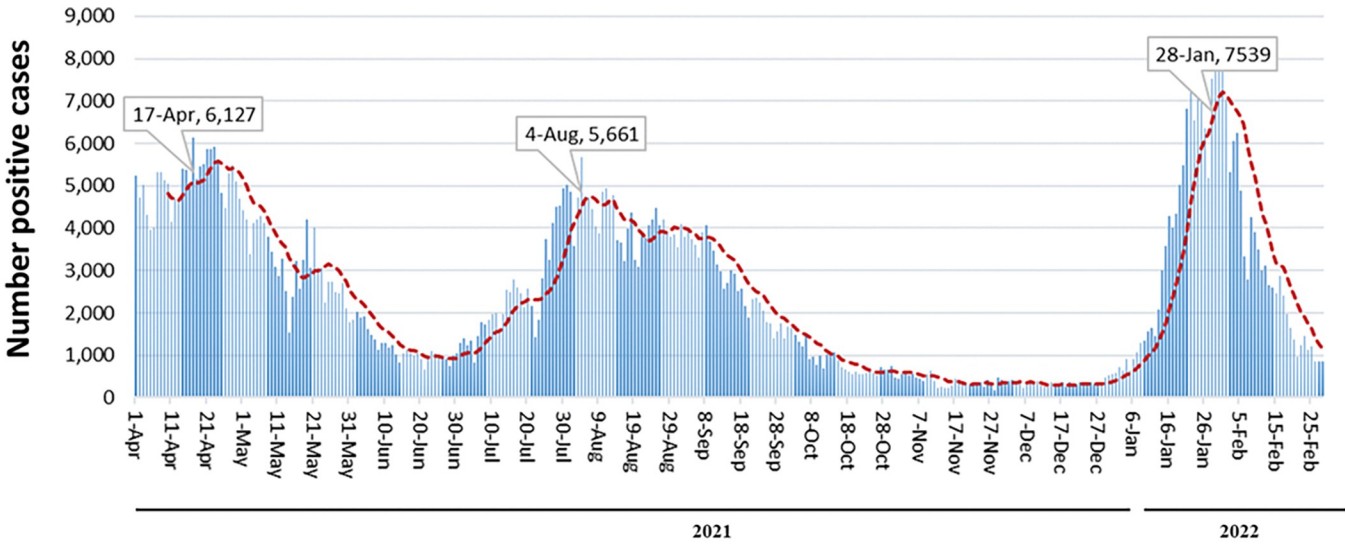

**Fig 1. SARS-CoV-2 PCR positive tests reported by the Aga Khan University Hospital Clinical Laboratories, Karachi between April 2021 and February 2022.** A, The graph depicts the number of SARS-CoV-2 PCR positive respiratory samples reported daily by the AKUH Clinical Laboratories, Karachi, Pakistan. The trend of daily positive cases is depicted with a red dotted trend line.

In April 2021, we used the GSD NovaType SARS-CoV-2 assay to screen for Alpha variants. The test differentiates between Alpha (N501Y and A570D) and Beta (N501Y) variants, (S1 and S2 Tables). Next, an updated version, GSD NovaType II SARS-CoV-2 assay that included (E484K and K417N) mutations became available. This was used to identify Alpha, Gamma, and Beta variants between May 2021 and December 2021. In May and June early Delta variants were identified from Pakistan [6], and PhoenixDx SARS-CoV-2 Mutant Screen [L452R] assay was employed. Shortages of this kit required us to use the NovaType Select P681R SARS-CoV-2 assays which identified Delta variant targets L452R and P681R, respectively between July and September 2021.

The NovaType III SARS-CoV-2 assay became available in August 2021 and was used until December 2021 for the identification of key lineage mutations for; Delta/Epsilon (E484, L452R), Kappa (E484Q, L452R) and Beta/Gamma (E484K) variants. By December 2021, the Omicron variant (BA.1) was identified [9], with its characteristic deletion in Spike codon 69–70 leading to a S- gene Target Failure (SGTF) identified on the TaqPath™ COVID-19 CE-IVD RT-PCR assay [10]. Thus, we employed the TaqPath assay until February 2022, keeping the NovaType III assay for screening of additional specimens.

## VOCs across the pandemic waves

We selected up to ten positive samples each day and screened 2150 (4% of) SARS-CoV-2 positive isolates tested at Aga Khan University Hospital Clinical Laboratories between April 2021 and April 2022 using the six above-mentioned RT-PCR assays. VOC were identified in 81.7% of samples tested (Fig 2). 18.3% of samples were 'undetermined' or could not be classified as a VOC based on the RT-PCR assays. We identified 4.2% (n = 90) Alpha, 7.8% (n = 168) Beta, 2.1% (n = 46) Gamma, 37.2% (n = 800) Delta, 1.8% (n = 39) Kappa and 17% (n = 367) Omicron variants amongst the VOC.

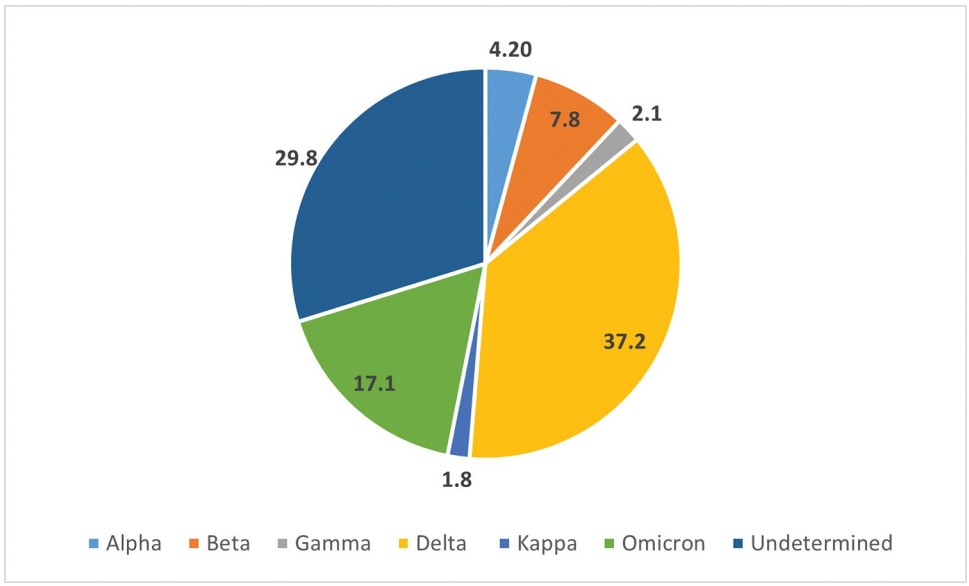

**Fig 2. SARS-CoV-2 variants in Karachi.** Data presented is of 2150 positive SARS-CoV-2 samples tested AKUH for Alpha, Beta, Gamma, Delta, Kappa and Omicron variants. The proportion of each VOC identified is shown by the pie chart for Alpha (n = 90), Beta (n = 168), Gamma (n = 46), Delta (n = 800), Kappa (n = 39) and Omicron (n = 367) variants. Samples which were none of these VOC are classified as Undetermined (n = 640).

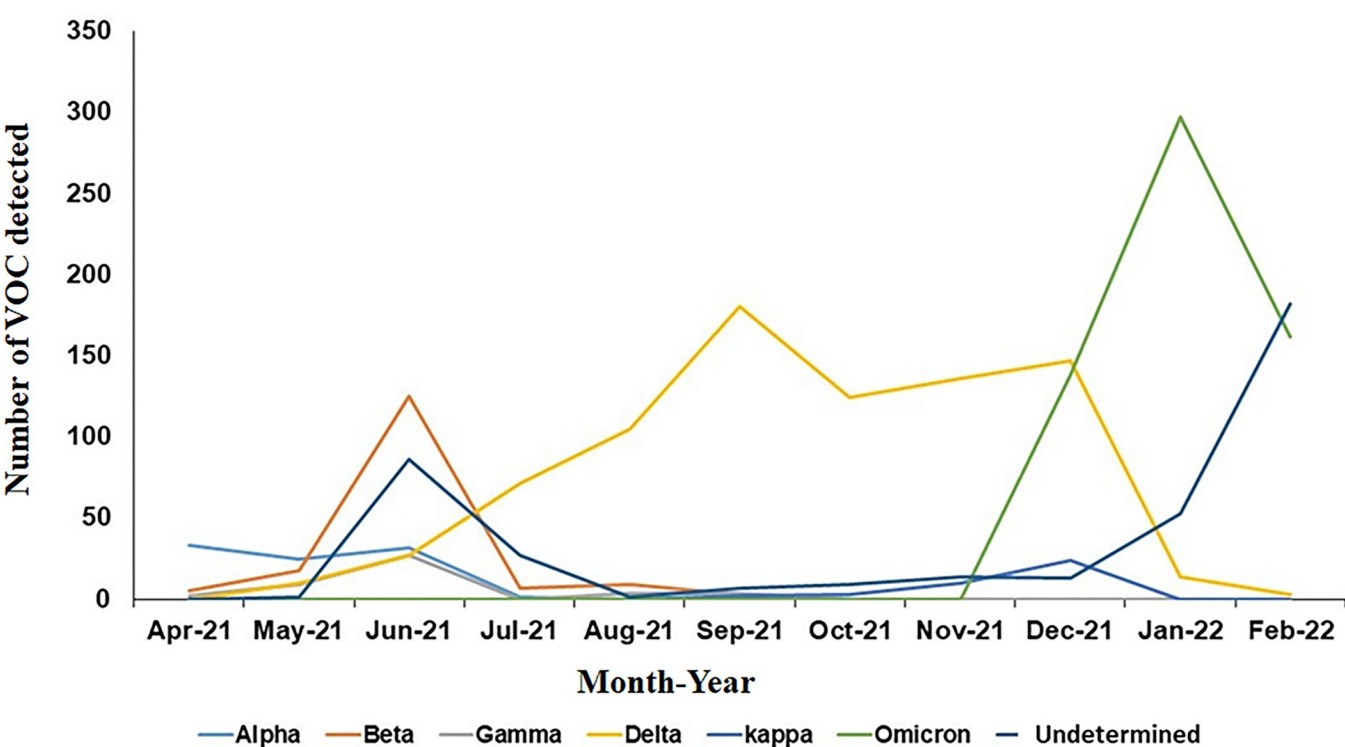

**Fig 3. The number of each VOC across the study period is shown.** The line graph depicts the number of each VOC; Alpha, Beta, Gamma, Delta, Kappa and Omicron together with undetermined samples which were identified across the study period.

The trend in which VOC were detected across the study period matches the COVID-19 wave observed (Fig 1). Alpha and Beta Variants were identified between April and June 2021 (Fig 3), and Gamma variants were observed albeit at a lower number through the same period. Delta variants first detected in May, became predominant through June, peaking in September and remaining prevalent until they were replaced by Omicron in December 2021. The highest number of VOC detected in a month was Omicron in January 2022 (297 cases) followed by Delta variants in September 2021 (180 cases), and Beta variants in June 2021 (125 cases).

The monthwise trend of Alpha variants were predominant in April and May 2021 at 82.5% and 40.3%, respectively (Fig 4). Beta variants increased in May (29%) and June (42%) and then reduced to 6% by July. Gamma variants were identified in May (14.5%) and June (9%), respectively. Delta variants first detected in May, comprised 66% of all variants by July, remaining dominant in August, September, October, and November 2021 at 88%, 91%, 91% and 85% of all cases respectively. Omicron (BA.1) variants emerged in December, rising to 42% of cases with an increase to 81% by January 2022 and then reducing to 45% in February 2022.

The variant of interest (VOI), Kappa was identified between September and December 2021, comprising between 1 and 7% of isolates tested within the months respectively. The proportion of 'Undetermined' samples varied, showing a peak at 29% in June 2021 and 52% in February 2022.

## Correlating VOC with COVID-19 severity

We compared the CT values of the samples in which VOC were identified (n = 1510) through the mutation screening. The CT values considered were the results of the SARS-CoV-2 diagnostic assay conducted for each clinical specimen. We observed a significant difference in CT

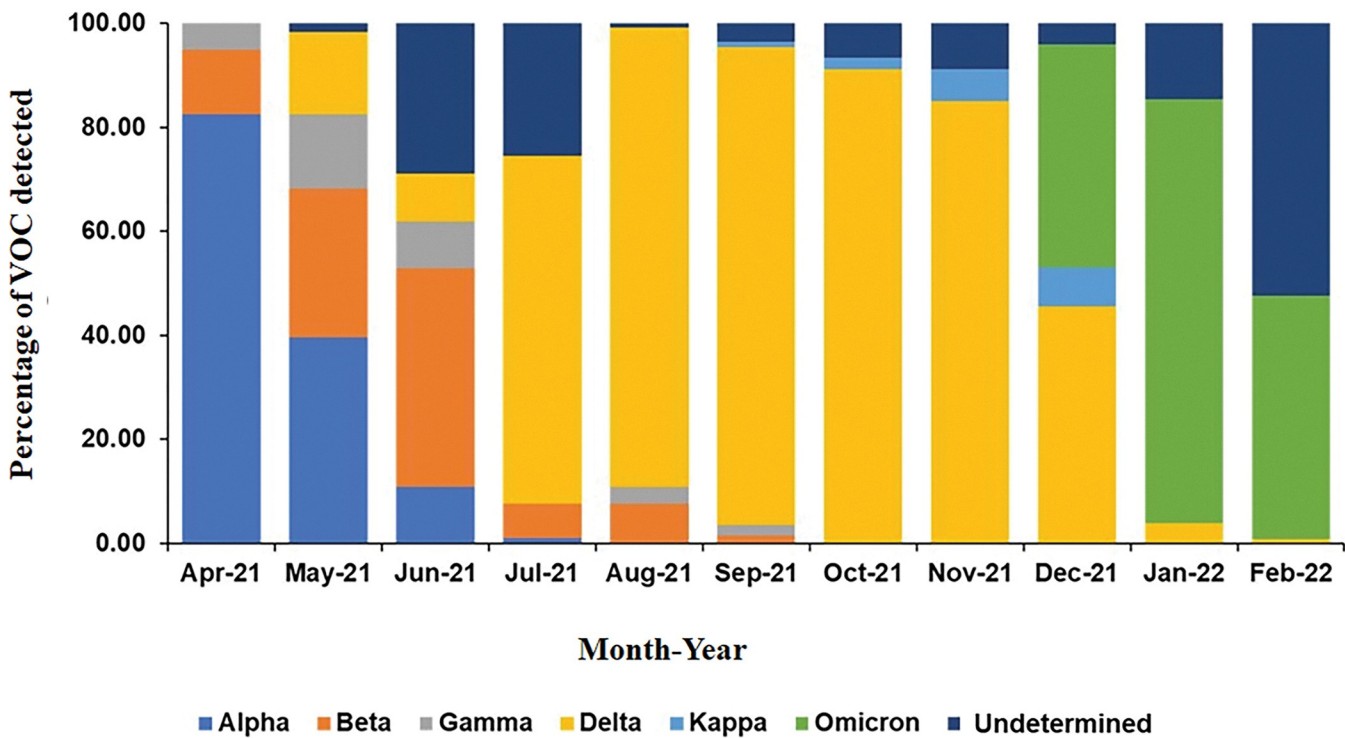

**Fig 4. The frequency of VOCs across the study period.** The graph depicts the frequency of each VOC; Alpha, Beta, Gamma, Delta, Kappa and Omicron together with undetermined samples which were identified across the study period.

values of the different VOC (Fig 5, p<0.0001). Further analysis revealed that CT values for Omicron were significantly lower as compared with Alpha (p<0.0001), Beta (p<0.0001), Gamma (p<0.0001) and Delta (p<0.0001) variants. Further CT values of Alpha were greater than for Beta (p = 0.0001) and Delta (p<0.0001) variants. SARS-CoV-2 viral loads have been shown to be associated with RT-PCR determined CT values of gene targets and are associated with COVID-19 severity [11, 12]. Therefore, it was apparent that Omicron variants had higher viral loads than Alpha, Beta, Gamma and Delta lineages. Notably, viral loads of samples with Alpha were lower than in the case of Beta and Delta variants.

Next, we examined the trend of VOCs across the study period in the context of COVID-19 cases (Fig 1) and associated mortality in Pakistan (S2 Table, using data from the COVID-19 Data Repository by the Center for Systems Science and Engineering (CSSE) at Johns Hopkins University. Mortality followed the increase in cases during waves in April and then between July 2021 and September 2021. The VOC during the former period was Alpha, with the Delta variant in the latter period. However, in the 2022 wave between January and February where, Omicron was predominant, a clear decoupling between COVID-19 cases and associated mortality was evident (S2 and S3 Figs).

## Discussion

We report the changing VOC associated with waves of the COVID19 pandemic in Pakistan. Genomic sequencing of SARS-CoV-2 strains can support infection control, epidemiological investigations and viral responses to vaccines and treatments. Sequencing efforts form Pakistan has contributed to the submission of 6,3583 SARS-CoV-2 genomes on GISAID, with 1,736 genomes available for April 2021 to February 2022 the period covered by this study

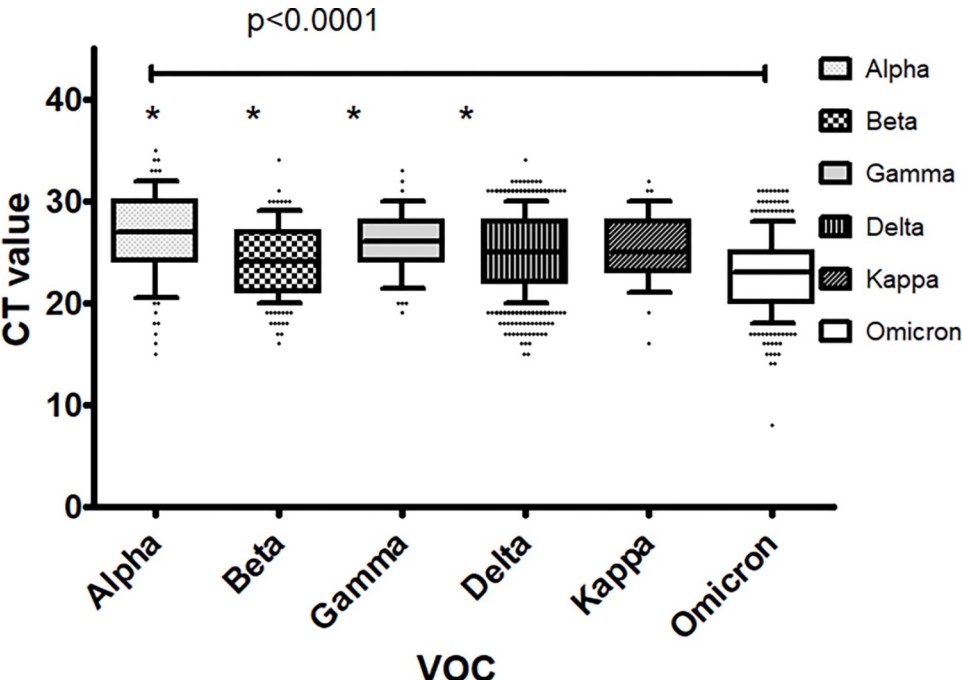

**Fig 5. Higher viral loads of Alpha, Beta and Delta as compared with Omicron variants.** The data represent the CT values of SARS-CoV-2 positive samples selected for VOC testing using specific mutation RT-PCR testing. CT values denote the results of the primary diagnostic SARS-CoV-2 PCR result on the cobas 6800 SARS-CoV-2 Roche assay (orf1ab gene target). Results for Alpha, Beta, Gamma, Delta, Kappa and Omicron variants are represented by a box and whiskers plots with IQR (10–90) and median shown by a horizontal line. Kruskal-Wallis analysis was run to determine significant differences between each VOC (p<0.05). '*', denotes p<0.05 in comparison with CT of Omicron variant using the Mann-Whitney U test.

(https://www.epicov.org/epi3/frontend#78ef7), link last accessed 10 April, 2023). WGS provides complete information regarding a pathogen, however it is costly to perform and difficult in a low resource setting. We used a targeted RT-PCR based approach to identify SARS-CoV-2 variants through the period April 2021 until February 2022.

The SARS-CoV-2 test data from AKUH Clinical Laboratories revealed three pandemic surges during the study period, classified as the third, fourth and fifth COVID-19 waves in Pakistan. Using the RT-PCR approach we demonstrated that Alpha and Beta variants were predominant in April 2021 and May 2021 respectively. Delta variants were first identified in May and became predominant from July until November 2021. The Omicron variant proportion rapidly increased in December 2021, associated with an increase in daily infections during the fifth wave of the pandemic.

Our data is from a single-center however, our COVID-19 data compares well with national trends. The Alpha variant was first introduced in Pakistan in December 2020 [8]. Beta and Gamma variants were identified in March and April 2021 [13]. Delta variants were present onwards of May and June 2021 [6]. The Omicron (BA.1) variant was introduced in December 2021 [14, 15].

The increase in Delta variant depicts the highly transmissible nature of the variant, supporting global news of its expansion in many countries such as, USA and UK [16, 17]. An advantage associated with Delta variants was the increased ability of spike protein to mediate inhibition of neutralizing antibodies compared with that of wild-type spike. This was further

enhanced in the case of Omicron which demonstrated increased transmissibility and the ability to evade antibody responses [18, 19].

Alpha, Beta, Gamma and Delta variants had CT values that were higher than those of the Omicron variants. SARS-CoV-2 viral loads have been inversely associated with CT values of RT-PCR assays [11, 12]. Therefore, it was apparent that viral loads were higher during the Omicron surge as compared with earlier VOC associated waves. Although higher viral loads have been associated with increased COVID-19 severity, there were lesser deaths in relation to the surge between December 2021 and February 2022 when Omicron was the predominant VOC. A decoupling of the number of cases and the mortality during the Omicron surge is documented [20, 21]. This reduced severity is likely a combination of increased existing immunity and vaccine coverage in the population [22]. Until June 2021, around 7,337,187 doses of the vaccine had been administered in Pakistan, however by December 2021 the number had significantly increased to 124,054,300 doses (https://covidvax.live/location/pak). The Omicron variant has shown high transmissibility but a lower CFR further, COVID-19 vaccination has also reduced disease severity and mortality [23].

One limitation is that RT-PCR testing does not fully characterize multiple genetic variations of SARS-CoV-2 lineages and study viral evolution. Furthermore, this method cannot identify minor and novel variants and several strains were 'undetermined'. Of note, we observed two-time intervals when the proportion of 'undetermined' cases was increased. This was, in May and June 2021 before the peak of Delta cases. Also, in January and February 2022, at the time of the shift from BA.1 to BA.2 Omicron variants. Therefore, a change in the pattern of 'undetermined' samples may indicate a transition between predominant VOCs.

Selected genomes were also sequenced as part of genomic surveillance and submitted in GISAID. We found the PCR-based VOC identification matched the Nextclade lineage-based categorization (S3 Table). Sequencing based identification also showed an 'undetermined' sample from January 2022, to be Omicron BA.2 variant.

The expansion of diagnostics during the COVID-19 pandemic has surpassed all previous expectations. However, during 2021, we faced difficulties in the import of supplies and reagents into Pakistan. The demand for viral extraction and testing reagents had grown enormously and supplies were not sufficient. Therefore, our usage of RT-PCR assays was based on their availability therefore, we changed the assay as required as it was a challenge to get a consistent supply kits. Also, we had to adapt to the emerging VOC. The assays used here were newly developed being built *in situ* in response to a new and growing need. We worked with local distributors who imported them for us based on their availability. Overall, as shown by other studies, Real-Time RT-PCR assays are an efficient tool for rapid detection and surveillance of VOC [24–26].

Scaling up of SARS-CoV-2 sequencing is limited by technical and financial capacity of countries. The sample size of our study was relatively small, but the sampling was consistent across the study period. Therefore, the trends in VOC we observed were similar to those observed elsewhere in Pakistan. Another advantage of the RT-PCR targeted mutation approach is that it allows testing of samples with viral loads up to CT 35. In contrast, WGS requires input RNA from samples of low CT values (high viral loads) preferably CT 25 and lower. Thus, WGS alone would result in an over-representation of VOCs with higher viral loads in the population. However, we observed that the RT-PCR test results were not consistent for samples CT >35 and are therefore not recommended for identifying VOC or samples with a very low viral load.

The data presented here highlights the value of linking lab and epidemiological data to the pandemic response, where it was helpful in informing the provincial and national health authorities supported by the WHO, Pakistan. This work lead to implementation of targeted

enforcement of COVID-19 Standard Operating Procedures and restrictions such as, quarantine, contact tracing and implementation of infection control practices. Identification of SARS-CoV-2 variant cases was reported on a weekly basis to the Health Department (NIH Situational Reports on the variants identified is available, https://www.nih.org.pk/novel-coronavirus-2019-ncov/), Sindh, Government of Pakistan, National Institutes of Health (Pakistan) and to WHO, Pakistan.

## Conclusions

An enormous global effort was required to curtail the spread of the SARS-CoV-2 virus worldwide. The establishment of the WHO's Technical Advisory Group on SARS-CoV-2 Virus Evolution (TAG-VE) and their bulletins would not have been possible without the efforts and contributions from affected countries, including Pakistan. SARS-CoV-2 continues to evolve, VOCs continue to emerge and their surveillance through lower-cost RT-PCR PCR based testing will be critical for assessing relatedness of viral strains within epidemiological clusters and supporting contact tracing and other public health interventions. In an epidemic setting it is valuable to have an adjunct approach where NGS based pathogen identification of key mutations associated with transmission may be traced using rapid, lower-cost PCR based testing.

## Supporting information

**S1 Fig. Workflow for how each RT-PCR kit was utilized to screen and differentiate the VOC.** The GSD NovaType SARS-CoV-2 assay was utilized to screen for Alpha and Beta variants. The test differentiates Alpha variant (N501Y and A570D) from Beta variant (N501Y). Subsequently, a newer version of the assay, GSD NovaType II SARS-CoV-2 assay was utilized to identify Alpha (N501Y), Gamma (E484K, N501Y) and Beta (N501Y, E484K, K417N) variants. Delta variants were also screened in parallel by using the assays PhoenixDx SARS-CoV-2 Mutant Screen [L452R] and NovaType Select P681R SARS-CoV-2 an assay that identified Delta variants by the L452R and P681R mutations. The NovaType III SARS-CoV-2 assay identified key lineage mutations for; Delta/Epsilon (E484, L452R), Kappa (E484Q, L452R) and Beta/Gamma (E484K) variants. The Omicron variant (BA.1) had been identified by utilizing TaqPath COVID-19 CE-IVD RT-PCR assay, omicron variants resulted in a S- gene Target Failure (SGTF).
(TIF)

**S2 Fig. COVID19 positive cases in Pakistan.** Data presented is for the COVID-19 positive cases in Pakistan between months April 2021 till February 2022. Source, John Hopkins, Corona Research Center https://coronavirus.jhu.edu/region/pakistan.
(TIF)

**S3 Fig. COVID19 related deaths in Pakistan.** Data presented is for the COVID-19 related deaths in Pakistan between months April 2021 till February 2022. Source, John Hopkins, Corona Research Center https://coronavirus.jhu.edu/region/pakistan.
(TIF)

**S1 Table. Timeline of SARS-CoV-2 variant testing kits employed.**
(DOCX)

**S2 Table. Details of SARS-CoV-2 variants testing PCR assays.** SARS-CoV-2 variants were identified with a PCR based approach targeting lineage specific mutations using commercially available assays. (A-F) present the specificity target of the probes/primers sequences and their

target mutation of these assays as provided by the manufacturer.
(DOCX)

**S3 Table. GISAID ID submissions as selected Alpha, Beta, Gamma, Delta, and Omicron variants.**
(XLSX)

## Acknowledgments

We thank for the following their support in this surveillance initiative; the AKUH Clinical Laboratory team particularly, Naima Maniar and Sohail Baloch. Thanks also to the COVID-19 testing staff at the Section of Molecular Pathology, AKUH.

## Author Contributions

**Data curation:** Akbar Kanji, Azra Samreen.

**Formal analysis:** Akbar Kanji, Ali Raza Bukhari, Zahra Hasan.

**Funding acquisition:** Zahra Hasan.

**Investigation:** Asghar Nasir, Uzma Bashir Aamir, Akbar Kanji, Zeeshan Ansar, Najia Karim Ghanchi, Kiran Iqbal Masood, Azra Samreen, Nazneen Islam, Samina Ghani, M. Asif Syed, Mansoor Wassan, Syed Faisal Mahmood.

**Methodology:** Asghar Nasir, Uzma Bashir Aamir, Akbar Kanji, Azra Samreen, Nazneen Islam, Zahra Hasan.

**Project administration:** Zahra Hasan.

**Resources:** Zeeshan Ansar, M. Asif Syed, Mansoor Wassan, Syed Faisal Mahmood, Zahra Hasan.

**Supervision:** Zahra Hasan.

**Validation:** Akbar Kanji, Azra Samreen.

**Visualization:** Akbar Kanji.

**Writing – original draft:** Asghar Nasir.

**Writing – review & editing:** Uzma Bashir Aamir, Ali Raza Bukhari, Zahra Hasan.

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
