## [Decision Letter · Decision Letter 0]

14 Mar 2023

PGPH-D-22-01797

Tracking SARS-CoV-2 variants through pandemic waves using PCR testing in low-resource settings

Dear Dr. Hasan,

Thank you for submitting your manuscript to PLOS Global Public Health. After careful consideration, we feel that it has merit but does not fully meet PLOS Global Public Health’s publication criteria as it currently stands. Therefore, we invite you to submit a revised version of the manuscript that addresses the points raised during the review process.

Please see comments from three reviewers below. The reviewers have requested some additional details in the methodology, and please ensure that SI Table 3 is clearly available, or that this reference is removed.

We look forward to receiving your revised manuscript.

Kind regards,

Hanna Landenmark

Staff Editor

Journal Requirements:

1. Our staff editors have determined that your manuscript is likely within the scope of our Diagnostics in Global Health Call for Papers. This editorial initiative is headed by a team of Guest Editors for PLOS GPH: Senjuti Saha (Child Health Research Foundation, Bangladesh) and Titus Divala (Public Health Scotland, University of Glasgow and University of Malawi College of Medicine). The Collection will encompass a diverse range of research articles about diagnostics in global health, including innovation and deployment of point of care diagnostics; subsets of diagnostics related to infectious diseases, chronic diseases and injuries; policies related to and regulation of diagnostics; supply chain issues; and the affordability, accessibility, and availability of essential diagnostics.  Additional information can be found on our announcement page: https://collections.plos.org/call-for-papers/diagnostics-in-global-health/

If you would like your manuscript to be considered for this collection, please let us know in your cover letter and we will ensure that your paper is treated as if you were responding to this call.  Please note that being considered for the Collection does not require additional peer review beyond the journal’s standard process and will not delay the publication of your manuscript if it is accepted by PLOS GPH. If you would prefer to remove your manuscript from collection consideration, please specify this in the cover letter.

2. Please amend your online detailed Financial Disclosure statement. This is published with the article. It must therefore be completed in full sentences and contain the exact wording you wish to be published.

a) State the initials, alongside each funding source, of each author to receive each grant. For example: "This work was supported by the National Institutes of Health (####### to AM; ###### to CJ) and the National Science Foundation (###### to AM)."

3. Please update your online Competing Interests statement. If you have no competing interests to declare, please state: “The authors have declared that no competing interests exist.”

4. In the online submission form, you indicated that your data will be submitted to a repository upon acceptance.  We strongly recommend all authors deposit their data before acceptance, as the process can be lengthy and hold up publication timelines. Please note that, though access restrictions are acceptable now, your entire data will need to be made freely accessible if your manuscript is accepted for publication. This policy applies to all data except where public deposition would breach compliance with the protocol approved by your research ethics board. If you are unable to adhere to our open data policy, please kindly revise your statement to explain your reasoning and we will seek the editor's input on an exemption. Please be assured that, once you have provided your new statement, the assessment of your exemption will not hold up the peer review process.

5. Please provide separate main figure files in .tif or .eps format only and ensure that all files are under our size limit of 10MB.

6. We do not publish any copyright or trademark symbols that usually accompany proprietary names, eg (R), (C), or TM  (e.g. next to drug or reagent names). Please remove all instances of trademark/copyright symbols throughout the text, including ® and ™ on pages 5 and 6 (cobas® and TaqPath™).

Additional Editor Comments (if provided):

Reviewers' comments:

Reviewer's Responses to Questions

**Comments to the Author**

1. Does this manuscript meet PLOS Global Public Health’s publication criteria? Is the manuscript technically sound, and do the data support the conclusions? The manuscript must describe methodologically and ethically rigorous research with conclusions that are appropriately drawn based on the data presented.

Reviewer #1: Yes

Reviewer #2: Yes

Reviewer #3: Partly

2. Has the statistical analysis been performed appropriately and rigorously?

Reviewer #1: Yes

Reviewer #2: N/A

Reviewer #3: No

3. Have the authors made all data underlying the findings in their manuscript fully available (please refer to the Data Availability Statement at the start of the manuscript PDF file)?

Reviewer #1: Yes

Reviewer #2: No

Reviewer #3: No

4. Is the manuscript presented in an intelligible fashion and written in standard English?

Reviewer #1: No

Reviewer #2: Yes

Reviewer #3: Yes

5. Review Comments to the Author

Reviewer #1: The topic is of interest, and the manuscript is well illustrated.

Major Comments:

1. Are there controversies in this field? What are the most recent and important achievements in the field? In my opinion, answers to these questions should be emphasized. Perhaps, in some cases, novelty of the recent achievements should be highlighted by indicating the year of publication in the text of the manuscript.

2. The results and discussion section is very weak and no emphasis is given on the discussion of the results like why certain effects are coming in to existence and what could be the possible reason behind them?

3. Conclusion: not properly written.

4. Results and conclusion: The section devoted to the explanation of the results suffers from the same problems revealed so far. Your storyline in the results section (and conclusion) is hard to follow. Moreover, the conclusions reached are really far from what one can infer from the empirical results.

5. The discussion should be rather organized around arguments avoiding simply describing details without providing much meaning. A real discussion should also link the findings of the study to theory and/or literature.

6. Spacing, punctuation marks, grammar, and spelling errors should be reviewed thoroughly. I found so many typos throughout the manuscript.

7. English is modest. Therefore, the authors need to improve their writing style. In addition, the whole manuscript needs to be checked by native English speakers.

Reviewer #2: The present study investigated circulating SARS-CoV-2 variants in Pakistan from 2020 to 2022. The authors determined the circulating variants of concern by detecting spike protein-specific mutations using a PCR-based technique and validated the results by sequencing.

In this study, the authors have identified and reported all five major variants of concern (Alpha, Beta, Gamma, Delta, and Omicron) through a rapid PCR-based approach. Their findings show that although the PCR-based technique cannot identify new circulating variants, the mutation-specific PCR-based technique is a rapid and cost-effective technique that can be used in the monitoring of known VOCs during pandemic waves in resource-limited settings.

The manuscript requires some revision.

• Results section, in screening for predominant VOCs there is some repetition of statements from the methodology, information such as primers, and mutations are already mentioned in the methods section. It is, therefore, better to remove the subsection to avoid redundancy.

• Supplementary Table 3 is not traceable in the manuscript file.

• Authors should show or provide the VOCs detection concordance % between the specific-mutation PCR technique versus NGS. Alternatively, provide sensitivity and specificity for the PCR- a technique in the detection of VOCs.

Reviewer #3: This manuscript describes a fast protocol to identify VOCs using RT-PCR instead of WGS, which may be expensive and more difficult. However, despite the practical relevance of the aim, some issues need to be carefully addressed.

1. The authors reported several kits employed for identifying various VOCs over time, but never mention previous literature (such as PMID: 35196812, PMID: 35204558, PMID: 35367360).On this subject, the discussion of studies with similar approach could also help for highlighting novel elements in the present studies. This part should be mentioned in the Introduction and then illustrated in the Discussion.

2. Furthermore, the manuscript does not fully describe the differences between the identified VOCs and more details on how each kit is able to identify the variants are needed. To this end, a suggestion would be to add a column to Supplementary Table 1 with the mutations related to each VOC.

3. Did the authors perform an evaluation of the limit of detection (in terms of viral load) for the variants during the related wave?

4. The authors should dwell more on the analysis and methods. More specifically, they should better describe the methods and the investigation that led them to make a correlation between VOC and COVID-19 severity in their case study. In particular, they should describe the employed statistical approach.

5. In the text, the authors very often mention PCR instead of RT-PCR. I suggest to use only RT-PCR for the sake of clarity.

6. I could not find the Supplementary table 3. On this subject, I reckon that in lines 238-241 the precise number of samples tested both by RT-PCR and WGS should be mentioned to make the study more reliable.

6. PLOS authors have the option to publish the peer review history of their article (what does this mean?). If published, this will include your full peer review and any attached files.

**Do you want your identity to be public for this peer review?** For information about this choice, including consent withdrawal, please see our Privacy Policy.

Reviewer #1: **Yes: **Talha Bin Emran

Reviewer #2: **Yes: **Doreen Donald Kamori

Reviewer #3: No

---

## [Decision Letter · Decision Letter 1]

5 May 2023

Tracking SARS-CoV-2 variants through pandemic waves using PCR testing in low-resource settings

PGPH-D-22-01797R1

Dear Dr. Hasan,

We are pleased to inform you that your manuscript 'Tracking SARS-CoV-2 variants through pandemic waves using PCR testing in low-resource settings' has been provisionally accepted for publication in PLOS Global Public Health.

Best regards,

Vijaykrishna Dhanasekaran, PhD

Academic Editor

Reviewer Comments (if any, and for reference):

Reviewer's Responses to Questions

**Comments to the Author**

1. If the authors have adequately addressed your comments raised in a previous round of review and you feel that this manuscript is now acceptable for publication, you may indicate that here to bypass the “Comments to the Author” section, enter your conflict of interest statement in the “Confidential to Editor” section, and submit your "Accept" recommendation.

Reviewer #1: All comments have been addressed

Reviewer #3: All comments have been addressed

2. Does this manuscript meet PLOS Global Public Health’s publication criteria? Is the manuscript technically sound, and do the data support the conclusions? The manuscript must describe methodologically and ethically rigorous research with conclusions that are appropriately drawn based on the data presented.

Reviewer #1: Yes

Reviewer #3: Yes

3. Has the statistical analysis been performed appropriately and rigorously?

Reviewer #1: Yes

Reviewer #3: N/A

4. Have the authors made all data underlying the findings in their manuscript fully available (please refer to the Data Availability Statement at the start of the manuscript PDF file)?

Reviewer #1: Yes

Reviewer #3: Yes

5. Is the manuscript presented in an intelligible fashion and written in standard English?

Reviewer #1: Yes

Reviewer #3: Yes

6. Review Comments to the Author

Reviewer #1: Results and conclusion: The section devoted to the explanation of the results suffers from the same problems revealed so far. Your storyline in the results section (and conclusion) is hard to follow. Moreover, the conclusions reached are really far from what one can infer from the empirical results.

Reviewer #3: The authors have sufficiently addressed the reviewers' comments.

7. PLOS authors have the option to publish the peer review history of their article (what does this mean?). If published, this will include your full peer review and any attached files.

**Do you want your identity to be public for this peer review?** For information about this choice, including consent withdrawal, please see our Privacy Policy.

Reviewer #1: **Yes: **Talha Bin Emran

Reviewer #3: No
